# Towards Skill and Population Curriculum for MARL

## Abstract

Recent advances in multi-agent reinforcement learning (MARL) allow agents to coordinate their behaviors in complex environments. However, common MARL algorithms still suffer from scalability and sparse reward issues. One promising approach to resolve them is *automated curriculum learning* (ACL), where *a student* (curriculum learner) train on tasks of increasing difficulty controlled by *a teacher* (curriculum generator). Unfortunately, in spite of its success, ACL's applicability is restricted due to: (1) lack of a general student framework to deal with the varying number of agents across tasks and the sparse reward problem, and (2) the non-stationarity in the teacher's task due to the ever-changing student strategies. As a remedy for ACL, we introduce a novel automatic curriculum learning framework, Curriculum Oriented Skills and Tactics (COST), adapting curriculum learning to multi-agent coordination. To be specific, we endow *the student* with population-invariant communication and a hierarchical skill set. Thus, the student can learn cooperation and behavior skills from distinct tasks with a varying number of agents. In addition, we model *the teacher* as a contextual bandit conditioned by student policies. As a result, a team of agents can change its size while retaining previously acquired skills. We also analyze the inherent non-stationarity of this multi-agent automatic curriculum teaching problem, and provide a corresponding regret bound. Empirical results show that our method improves scalability, sample efficiency, and generalization in MPE and Google Research Football. The source code and the video can be found at https://sites.google.com/view/neurips2022-cost/.

## 1 Introduction

Multi-agent Reinforcement Learning (MARL) has long been a go-to tool in complex robotic and strategic domains [1, 2]. However, learning effective policies with sparse reward from scratch for large-scale multi-agent systems remains challenging. One of the challenges is that the joint observation-action space grows exponentially with varying numbers of agents. Meanwhile, the sparse reward signal requires a large number of training trajectories. Hence, applying existing MARL algorithms directly to complex environments with a large number of agents is not effective. In fact, they may produce agents that do not collaborate with each other even when it is of significant benefit [3, 4].

There are several lines of work related to the large-scale MARL problem with sparse reward, including: reward shaping [5], curriculum learning [6], and learning from demonstrations [7]. Among these approaches, the curriculum learning paradigm, in which the difficulty of experienced tasks and the population of training agents progressively grow, shows particular promise. In *automatic* curriculum learning (ACL), a teacher (curriculum generator) learns to adjust the complexity and sequencing of tasks faced by a student (curriculum learner). Several works have even proposed *multi-agent* ACL algorithms, based on approximate or heuristic approaches to teaching, such as DyMA-CL [8], EPC

[9], and VACL [6]. However, DyMA-CL and EPC rely on a framework of an off-policy student with replay buffer, and ignore the forgetting problem that arises when the agent population size grows. In turn, VACL relies on the strong assumption that the value of the learned policy does not change when agents switch to a different task. Moreover, the teacher in these approaches is still facing an unmitigated non-stationarity problem due to the ever-changing student strategies. In addition, if we somewhat expand the ACL paradigm and presume that the teacher may have another purpose for the sequence of tasks performed by the student, another class of larger-scale MARL solutions should be mentioned. Namely, hierarchical MARL, which learns temporal abstraction with more dense rewards, including: skill discovery [10], option as response [11], role-based MARL [12], and two levels of abstraction [13]. Alas, hierarchical MARL mostly focuses on one specific task with a fixed number of agents and does not consider the transfer ability of learned complementary skills. Interestingly, as we show in this paper, a smart merger of ACL and hierarchical MARL principles can overcome their combined weaknesses and more.

Specifically, in this paper, we introduce a novel automatic curriculum learning algorithm, Curriculum Oriented Skills and Tactics (COST), which learns cooperative behaviors from scratch. The core idea of COST is to encourage the student to learn skills from tasks with different parameters and different numbers of agents. Motivation from the real world is team sports, where players often train their skills by gradually increasing the difficulty of tasks and the number of coordinating players. In particular, we implement COST with three key components. First, to handle the varying number of agents across tasks, motivated by the transformer [14], which can process sentences of varying lengths, we implement population-invariant communication by treating each agent's message as a word. Thus, a self-attention communication channel is used to support an arbitrary number of agents sharing their messages. Second, to learn transferable skills in the sparse reward setting, we utilize the skill framework in the student. Agents communicate on the high level about a set of shared low-level policies. Third, to address the non-stationarity arising from ever-changing student strategies, we model the teacher as a contextual bandit, where we utilize an RNN-based [15] imitation model to represent student policies and use this to generate the bandit's context. Empirical results show that our method achieves state-of-the-art performance in several tasks in the multi-particle environment (MPE) [16] and the challenging 5vs5 competition in Google Research Football [17].

## 2 Further Related Work

**(Automatic) Curriculum Learning in (MA)RL.** Curriculum learning is a training strategy inspired by the human learning process, mimicking how humans learn new concepts in an orderly manner, usually based on the difficulty level of the problems [18]. The selection of tasks is formulated as a Curriculum Markov Decision Process (CMDP) [19]. Automatic Curriculum Learning mechanisms aim to learn a task selection function based on information about past interactions, such as ADR [20, 21], ALP-GMM [22], SPCL [23] and GoalGAN [24]. Inspired by the mechanism of biodiversity in nature, a series of MARL curriculum learning frameworks have recently been proposed with remarkable empirical success. These include open-ended evolution [25–27], population-based training [28, 29], and training with emergent curriculum [18, 30, 31]. In general, these frameworks can be unified under the idea of an automatic curriculum that automatically generates an endless procession of better performing agents by exerting selection pressure among many self-optimizing agents.

**Hierarchical MARL and Communication.** Hierarchical reinforcement learning (HRL) has been extensively studied to address the sparse reward problem and to facilitate transfer learning. Single-agent HRL focuses on learning the temporal decomposition of tasks, either by learning subgoals [32–37] or by discovering reusable skills [38–41]. Recent works about hierarchical MARL have been discussed in the Introduction. In multi-agent settings, communication has demonstrated success in multi-agent cooperation [42–48]. However, existing approaches that extend HRL to multi-agent systems or utilize communication are limited to a fixed number of agents and are hard to transfer with different number of agents.

**Multi-armed Bandit.** Multi-armed bandits (MABs) are a simple but very powerful framework that repeatedly makes decisions under uncertainty. In an MAB, a learner performs a sequence of actions. After every action, the learner immediately observes the reward corresponding to its action. Given a set of $K$ actions and a time horizon $T$, the objective is to maximize its total reward over $T$ rounds. The regret is used to measure the gap between the cumulative reward of an MAB algorithm and the

best-arm benchmark. A related work is the Exp3 algorithm [49], which is proposed to increase the probability of pulling good arms and achieves a regret of $O(\sqrt{KT\log(K)})$ under a time-varying reward distribution. Another related work is the contextual bandit problem [50], where the learner makes decisions based on prior information. In this work, the teacher is modeled as a contexual bandit. We learn the dynamic context, leverage the Lipschitz assumption with respect to the context, and provide a regret bound of the proposed method.

**Google Research Football [17].** There are some challenges in the GRF (see Fig. 2). (1) Large-scale problem: In the GRF, for cooperative players, the joint action space is large; therefore, it is difficult to build a single agent to control all players. Moreover, the opponents are not fixed due to a stochastic environment and a difficulty configuration, and the agents should be adapted to various opponents. (2) Sparse rewards: The goal of the football game is to maximize the scores, which can only be obtained after a long time by iteration. Therefore, it is almost impossible to receive a positive reward when starting with random agents. Recent works attempt to tackle multi-agent scenarios in GRF by using a containerized learning framework [51], learning from demonstration [7], individuality [52], and diversity [53]. However, they mainly focus on single-agent control, or train relatively easy academy tasks in GRF, or use offline expert data to train agents.

# 3   Problem Formulation: MARL with Curriculum

**Dec-POMDP.** An MARL problem is formulated as a *decentralised partially observable Markov decision process* (Dec-POMDP) [54], which is described as a tuple $\langle n, \boldsymbol{S}, \boldsymbol{A}, P, R, \boldsymbol{O}, \boldsymbol{\Omega}, \gamma \rangle$, where $n$ represents the number of agents. $\boldsymbol{S}$ represents the space of global states. $\boldsymbol{A} = \{A_i\}_{i=1,\cdots,n}$ denotes the space of actions of all agents. $\boldsymbol{O} = \{O_i\}_{i=1,\cdots,n}$ denotes the space of observations of all agents. $P : \boldsymbol{S} \times \boldsymbol{A} \to \boldsymbol{S}$ denotes the state transition probability function. All agents share the same reward as a function of the states and actions of the agents $R : \boldsymbol{S} \times \boldsymbol{A} \to \mathbb{R}$. Each agent $i$ receives a private observation $o_i \in O_i$ according to the observation function $\boldsymbol{\Omega}(s, i) : \boldsymbol{S} \to O_i$. $\gamma \in [0, 1]$ denotes the discount factor.

**Curriculum-enhanced Dec-POMDP.** A Dec-POMDP is defined by a tuple $\langle \Phi, \mathcal{M} \rangle$ where $\Phi$ is the task space. Given a task $\phi$, a Dec-POMDP $\mathcal{M}(\phi)$ is presented as $\left\{ n^\phi, \boldsymbol{S}^\phi, \boldsymbol{A}^\phi, P^\phi, r^\phi, O^\phi, \boldsymbol{\Omega}^\phi, \gamma^\phi \right\}$. The superscript $\phi$ denotes that the Dec-POMDP elements are determined by the task $\phi$. Note that task $\phi$ can be a few parameters of the environment or task IDs in a finite task space. *In a curriculum-enhanced Dec-POMDP, the objective is to improve the student's performance on the target tasks by the teacher's giving the sequence of training tasks.*

Let $\tau$ denote a trajectory whose unconditional distribution $\mathrm{Pr}_\mu^{\pi,\phi}(\tau)$ under a policy $\pi$ and a task $\phi$ with initial state distribution $\mu(s_0)$ is $\mathrm{Pr}_\mu^{\pi,\phi}(\tau) = \mu(s_0) \sum_{t=0}^{\infty} \pi(a_t \mid s_t) P^\phi(s_{t+1} \mid s_t, a_t)$. We use $p(\phi)$ to represent the distribution of target tasks and $q(\phi)$ to represent the distribution of training tasks at each task sampling step. Considering the joint agents' policies $\pi_\theta(a|s)$ and $q_\psi(\phi)$ parameterized by $\theta$ and $\psi$, respectively. The overall objective to maximize in a curriculum-enhanced Dec-POMDP is:

$$J(\theta, \psi) = \mathbb{E}_{\phi \sim p(\phi), \tau \sim \mathrm{Pr}_\mu^\pi} \left[ R^\phi(\tau) \right] = \mathbb{E}_{\phi \sim q_\psi(\phi)} \left[ \frac{p(\phi)}{q_\psi(\phi)} V(\phi, \pi_\theta) \right] \tag{1}$$

where $R^\phi(\tau) = \sum_t \gamma^t r^\phi(s_t, a_t; s_0)$ and $V(\phi, \pi_\theta)$ represent the value function of $\pi_\theta$ in Dec-POMDP $\mathcal{M}(\phi)$. However, when optimizing $q_\psi(\phi)$, we cannot get the partial derivative $\nabla_\psi J(\theta, \psi) = \nabla_\psi \sum_\tau \frac{1}{q_\psi(\phi)} R^\phi(\tau) \mathrm{Pr}_\mu^{\pi,\phi}(\tau)$[1] since the reward function and the transition probability function w.r.t number of agents are non-parametric, non-differentiable, and discontinuous in most MARL scenarios.

Thus, we use the non-differentiable method, i.e., multi-armed bandit algorithms to optimize $q_\psi(\phi)$, and optimize the overall objective by learning the distribution of training tasks (the teacher) and an RL algorithm (the student) in alternating periods. However, there are three key challenges in solving this problem: (1) There is a lack of a general student framework to deal with the varying number of agents across tasks and the sparse reward problem. (2) The teacher is facing a non-stationarity problem due to the ever-changing student's strategies. (3) The forgetting and relearning problem.

---

[1]$p(\phi)$ is not in the partial derivative since it is a fixed distribution.

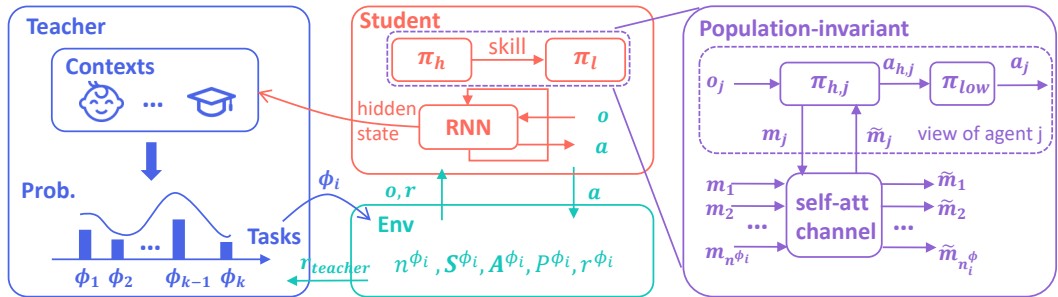

Figure 1: The overall framework of COST. COST is composed of three parts: configurable environments, a teacher, and a student. **Left.** The teacher is modeled as a contextual multi-armed bandit. At each teacher timestep, the teacher chooses a training task from the distribution of bandit actions. **Mid.** The student is endowed with population-invariant communication and a skill framework, and trained with MARL algorithms on the training task. The student returns to the teacher not only the hidden state of RNN imitation model as contexts but also the average discounted cumulative rewards on the testing task. **Right.** The student learns hierarchical policies. The population-invariant communication is on the high level, and implemented with a self-attention communication channel to handle the messages from varying number of agents. The agents in the student share the same low-level policy.

Some tasks can be the prerequisites of other tasks and some tasks can be inter-independent and parallel. For tackling these challenges, in the following section, we propose a novel multi-agent automatic curriculum learning framework, Curriculum Oriented Skills and Tactics (COST).

# 4 Curriculum Oriented Skills and Tactics

In this section, we present our automatic curriculum learning algorithm named Curriculum Oriented Skills and Tactics (COST) as shown in Fig. 1. First, we present the student with a skill and population-invariant communication framework to tackle the varying number of agents and the sparse reward problem. Then, to deal with the non-stationarity as well as unknown prior knowledge, we propose a contextual multi-armed bandit algorithm as the teacher.

## 4.1 Student with Population-invariant Communication and Skills

In the student, we treat many agents as a whole and apply the MARL algorithms to train the student. To address the varying number of agents, we propose a population-invariant communication framework where agents can communicate via a self-attention channel. Moreover, to deal with the sparse reward problem, we introduce a skill framework in which agents can learn the skills (high-level actions) that can be transferred among different tasks.

**Population-invariant Communication.** Instead of learning independent policies for agents in the student, we introduce communication to enable the population-invariant property and learn tactics among agents. Motivated by the fact that the transformer [14] in natural language processing can handle varying lengths of sentences, we use the self-attention mechanism in our communication. As shown in Fig. 1 Right, each agent $j$ receives an observation $o_j$. In each round of communication, each agent $j$ sends a message vector $m_j = f(o_j)$ to a self-attention channel, where $f$ is an observation encoder function.

The channel aggregates all messages and sends the new message vector $\tilde{m}_j$ through the self-attention mechanism. Concretely, given the input of the channel $\mathbf{M} = [m_1, m_2, \cdots, m_n] \in R^{n \times d_m}$ and the trainable weight of the channel $\mathbf{W}_Q, \mathbf{W}_K, \mathbf{W}_V \in R^{d_m \times d_m}$, we can obtain three different representations $\mathbf{Q} = \mathbf{M}\mathbf{W}_Q, \mathbf{K} = \mathbf{M}\mathbf{W}_K, \mathbf{V} = \mathbf{M}\mathbf{W}_V$. Then, the output messages are

$$\tilde{\mathbf{M}} = \text{Attention}(\mathbf{Q}, \mathbf{K}, \mathbf{V}) = \text{softmax}\left(\frac{\mathbf{Q}\mathbf{K}^T}{\sqrt{d_m}}\right)\mathbf{V} = [\tilde{m}_1, \tilde{m}_2, \cdots, \tilde{m}_n] \quad (2)$$

where $d_m$ is the dimension of the messages. Since the dimensions of the trainable weight are irrelevant to the number of agents, the student can take advantage of the population-invariant property to learn tactics.

**Skill Framework in Student.** As shown in the dotted box in Fig. 1 Right, after receiving the new messages $\tilde{m}_j$ from the channel, each agent takes the high-level action (skill) $a_{h,j} = \pi_{h,j}(o_j, \tilde{m}_j)$ to execute the low-level policy $a_j = \pi_{low}(o_j, a_{h,j})$. In this work, we generalize the high-level action (skill) $a_{h,j}$ to a continuous embedding space, so that the skill can be either a latent continuous vector as in DIAYN [55], or a categorical distribution for sampling discrete options [56].

We implement the high- and low-level policies in the student with PPO [57]. The high-level policy for each agent is learned independently, whereas the low-level policies share parameters, since the most basic action pattern should be the same within different agents. The low-level agent is rewarded by the environment. The high-level policy takes actions given a fixed interval during training. Within this interval, a cumulative low-level reward is used as a high-level reward. When the categorical distribution is used to enable an option-style skill, we would sample an "option" from the categorical distribution and feed the corresponding one-hot embedding to the low-level policy.

### 4.2 Teacher: Contextual Bandit in a Non-stationary Environment

The teacher is expected to guide the student to learn the skills and tactics by offering and ordering different tasks. However, since the student learns across different tasks, the teacher is facing a non-stationarity problem due to the ever-changing student's strategies. That is, in different stages of student learning, the teacher will observe different student's performances when giving the same task to the student, thus leading to a time-varying reward distribution of the teacher.

In addition, there exists the forgetting and relearning problem of the student, where the student forgets the learned policy. To avoid this problem, the teacher should offer some trained tasks to the student. It can be seen as the exploitation and exploration problem of the teacher. The teacher is encouraged to give the training tasks that benefit the student's performance on the target tasks; however, there is still a need for sufficient exploration on various training tasks.

Fortunately, we notice that the non-stationarity stems from the student, which can be mitigated with a contextual bandit which embeds the student policy into the context. As shown in Fig. 1 Left, the teacher takes the student's policy representation as the context and chooses a task from the distribution of training tasks. Specifically, we extend the Exp3 algorithm [49] with context by utilizing an online cluster algorithm BIRCH [58] in Alg.1. The context $x$ is the student's policy representation, the teacher's action is a certain task $\phi$, and the teacher's reward is the return of the student in the target tasks. In steps 1-4, the teacher samples a task for the student's training, and in steps 6-7, the teacher would update the parameters based on the evaluation reward of the student.

---

**Algorithm 1** Teacher Sampling and Training

---

**Input:** Context $x$, the number of Clusters $N_c$, $N_c$ instances of Exp3 with task distribution $w(\phi_k, c)$ for $k = 1, \ldots, K$ and for $c = 1, \ldots, N_c$, learning rate $\alpha$, a buffer maintaining the historical contexts

**Output:** $\mathcal{M}(\phi) = \left\{ n^\phi, \mathbf{S}^\phi, \mathbf{A}^\phi, P^\phi, r^\phi, O^\phi, \mathbf{\Omega}^\phi, \gamma^\phi \right\}$, the teacher bandit parameters

**Sampling**

1. Get the the context $x$, and save it to the buffer
2. Run the online cluster algorithm and get the index of the cluster center $c(x)$
3. Let the active Exp3 instance be the instance with index $c(x)$
4. Set the probability $p(\phi_k, c(x)) = \frac{(1-\alpha)w(\phi_k, c(x))}{\sum_{j=1}^{K} w(\phi_k, c(x))} + \frac{\alpha}{K}$ for each task $\phi_k$
5. Sample a new task according to the distribution of $p_{\phi_k, c}$

**Training**

6. Get the return (discounted cumulative rewards) from student testing $r$
7. Update the active Exp3 instance by setting $w(\phi_k, c(x)) = w(\phi_k, c(x))e^{\alpha r/K}$

---

### 4.2.1 Context Representation

We learn the representation of the student policy as a context. A straightforward representation is to directly use the student parameters $\theta$ as the context. However, the number of parameters is large and ever-changing if we change the student's architecture. Thus, we turn to an alternative method.

A principle to learn a good representation of a policy is *predictive representation*, that is, the representation should be accurate to predict policy actions given states. According to the principle, we utilize an imitation function through supervised learning. Supervised learning does not require direct access to reward signals, making it an attractive task for reward-agnostic representation learning. Intuitively, the imitation function attempts to mimic low-level policy based on historical behaviors. In practice, we use an RNN-based imitation function $f_{im} : \mathcal{S} \times \mathcal{A} \to [0, 1]$. Since recurrent neural networks are theoretically Turing complete [59], its internal states can be used as the representation of the student's policy. Regarding the training of this imitation function, we use the negative cross entropy objective $\mathbb{E}[\log f_{im}(s, a)]$.

### 4.2.2 Regret Analysis

In this subsection, we show the regret bound of the proposed teacher algorithm $\mathbb{E}[R(T)] = O\left(T^{2/3}(LK \log T)^{1/3}\right)$, where $T$ is the number of total rounds, $L$ is the Lipschitz constant, and $K$ is the number of arms (the number of the teacher's actions). Since the teacher's reward is the return of the student in the target tasks, the regret bound shows the optimality of the proposed method.

First, we introduce the Lipschitz assumption about the generalization ability of the task space.

**Assumption 4.1** (Lipschitz continuity w.r.t the context)**.** Without loss of generality, the contexts are mapped into the $[0, 1]$ interval, so that the expected rewards for the teacher are Lipschitz with respect to the context.

$$|r(\phi \mid x) - r(\phi \mid x')| \leq L \cdot |x - x'|$$

for any arm $\phi \in \Phi$ and any pair of contexts $x, x' \in \mathcal{X}$

(3)

where $L$ is the Lipschitz constant, and $\mathcal{X}$ is the context space.

This assumption suggests that, for any policy that is trained on a set of tasks, the rate of performance change is not faster than the rate of policy change. It is a realistic assumption since we cannot expect the student to achieve a dramatic improvement on a given task when the student is represented by a new context via a few training steps.

Then, we borrow a contextual bandit algorithm for a small number of contexts [49] (see Appendix Alg. 2) and the lemma 4.2, as a stepping stone for the proof of Theorem 4.3.

**Lemma 4.2.** *Alg. 2 has regret* $E[R(T)] = \mathcal{O}(\sqrt{TK|\mathcal{X}| \log K})$.

Lemma 4.2 introduces a square root dependence on $|\mathcal{X}|$ if running a separate copy of Exp3 for each context [49]. It motivates us to handle large context space by discretization.

**Theorem 4.3.** *Consider the Lipschitz contextual bandit problem with contexts in* $[0, 1]$. *The Alg. 1 yields regret* $\mathbb{E}[R(T)] = O\left(T^{2/3}(LK \ln T)^{1/3}\right)$.

*Proof.* See Appendix B for the proof. □

In practice, the contextual space is high-dimensional instead of in $[0, 1]$, and in the proof a uniform mesh is used to discretize the context space. Since we cannot have such a uniform mesh, without loss of generality, we use the BIRCH streaming data cluster algorithm [58] to generate and discretize the context space. At the end of the training, the cluster can be seen as an approximation of the uniform mesh.

## 5 Experiments

We consider several tasks in two environments, Simple-Spread and Push-Ball in the Multi-agent Particle-world Environment (MPE) [16], and the challenging 5vs5 task of GRF [17], to further demonstrate the performance of our approach.

We aim to answer the following three research questions. **Q1**: *Is curriculum learning needed in the complex large-scale MARL problem?* (See Sec. 5.2) **Q2**: *Can our COST outperform previous curriculum-based MARL methods? If so, which components in COST contributes the most to performance gains?* (See Sec. 5.3) **Q3**: *Can COST learn a good curriculum for the student?* (See Sec. 5.4)

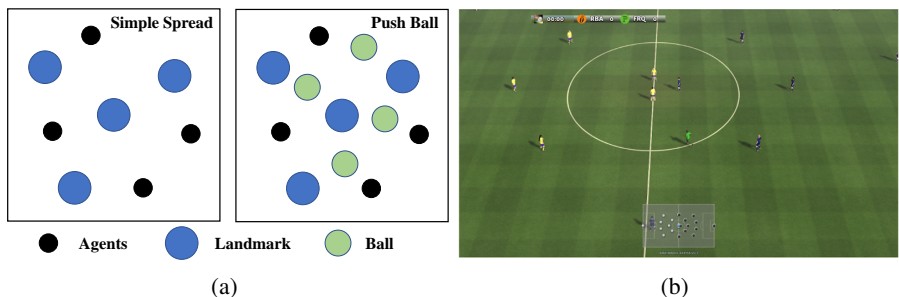

Figure 2: The environments. (a): Multi-particle Environment. (b): Google Reaserach Football

## 5.1 Environments, Baselines and Metric

**Environments.** In the GRF 5vs5 scenario, we need to control 4 agents (except the goalkeeper) to compete with the opponent built-in AI. Each agent would observe a compact encoding, which consists of a 115-dimensional vector summarizing many aspects of the game, such as player coordinates, ball possession and direction, active player, and game mode. The action set available to an individual agent consists of 19 discrete actions such as idle, movement, passing, shooting, dribbling, or sliding. The GRF provides two types of reward: scoring and checkpoints, to encourage the agent to move the ball forward and have a successful shot.

Table 1: Baseline algorithms.

| Categories | Methods |
|---|---|
| MARL (**Q1**) | QMIX [60] 
 IPPO [61] |
| Curriculum-based (**Q2**) | IPPO with uniform task sampling 
 VACL [6] |
| Ablation Study (**Q3**) | COST with uniform task sampling 
 COST without HRL |

In MPE, we investigate `Simple-Spread` and `Push-Ball` (see Fig. 2a). In `Simple-Spread`, there are $n$ agents that need to cover all $n$ landmarks. Agents are penalized for collisions and only receive a positive reward when all the landmarks are covered. In `Push-Ball`, there are $n$ agents, $n$ balls, and $n$ landmarks. The agents need to push the balls to cover every landmark. A success reward is given after all the landmarks have been covered.

**Baselines.** We evaluate the following approaches as baseline in Table 1:

We compare MARL algorithms to justify curriculum learning in the complex large-scale MARL problem. Also, we modify VACL by removing the centralized critic for a fair comparison of the MPE. Due to the difficulty of the GRF, we include a shooting reward to encourage the student to shoot.

**Metric.** Even if we use the reward to optimize various algorithms, the mean episode reward in such environments cannot show the performance of the agents. Therefore, for GRF scenarios, we plot the win rate and the average goal difference, which is the number of goals scored by the MARL agents minus the number of goals scored by the other team.

The experiments are carried out on 30 nodes, one of which has a 128-core CPU and 4 A100 GPUs. Each experiment trial is repeated over 5 seeds and runs for 1-2 days.

## 5.2 The Necessity of Curriculum Learning

First, we describe experiments using MPE. In contrast to the fully observable setting and the centralized critic in VACL, we consider individual PPO in partially observable environments with default rewards. We randomly pick a starting state, and the episode ends after a fixed number of maximum steps. To be specific, the task space consists of $n$ agents, where $n \in \{2, 4, 8, 16\}$. We set the maximum allowed steps to 25. All evaluations are performed on the target task, where $n = 16$. IPPO is trained and evaluated directly on the target task. In Fig. 3, we can see that IPPO performs nearly VACL. COST achieves a higher coverage rate than the baseline methods, but the improvement

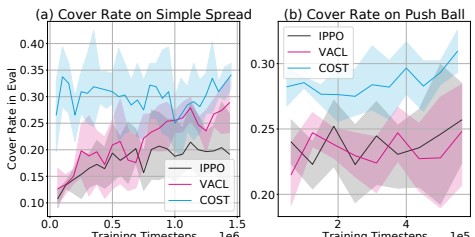
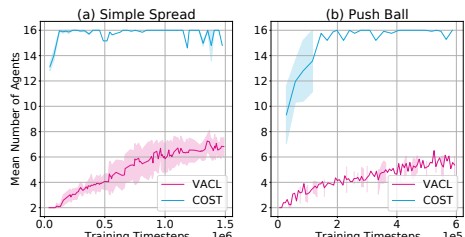

Figure 3: The evaluation performance of various methods on MPE.

Figure 4: The changes in the number of agents on MPE.

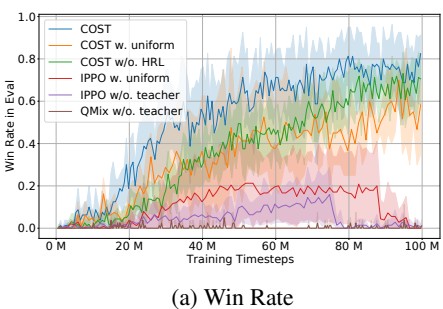

(a) Win Rate

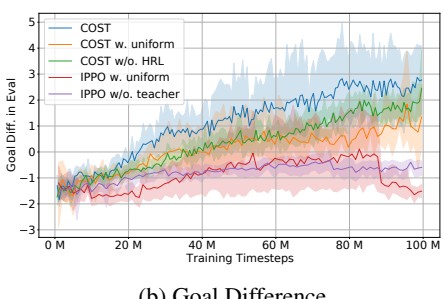

(b) Goal Difference

Figure 5: The evaluation performance of various methods on 5vs5 football competition.

is not significant. Furthermore, we experimentally investigate the probability variation of different population sizes in Fig. 4. We observe that the curriculum afforded by COST is approaching the target task. The results illustrate that in a simple environment where the student can directly learn to complete the task, there is no need to apply curriculum learning.

Then we show the performance comparison with the baselines in GRF. We also run CDS [53] and CMARL [51], however, we did not include their performances, since the goal difference reported in CMARL [51] is relatively low compared to our method. In Fig. 6, we can see that without the curriculum learning scheme, QMix and IPPO cannot perform well in the 5vs5 scenario. However, IPPO is slightly better than QMix in the scope of MARL algorithms in this scenario. In Fig. 5b, we omit the lines of QMix since the mean score is low, affecting the presentation of the figure. The reason could be that QMix is an off-policy MARL algorithm, which would rely heavily on the replay buffer. However, in such sparse reward scenarios, the replay buffer has much less efficient samples for QMix to learn. For example, the replay buffer would contain tons of zero-score samples, leading to a non-promising performance. Meanwhile, IPPO with a shared actor and critic, an on-policy algorithm, would utilize the samples more efficiently. Therefore, curriculum learning is a promising solution to the complex large-scale MARL problem.

During our experiments, we found that IPPO or shared parameter PPO can easily achieve good performance in most academic scenarios in GRF. However, 5vs5 is an obstacle for agents to handle more complex scenarios. Due to the limitation of computational resources, we tested COST in the 11vs11 scenario. The result can be seen in the Appendix C.

**5.3 Performance and Ablation Study**

In the experiments on MPE, In both environments, COST performs better than VACL. Instead of training with continuous relaxation of the categorical distribution of population size in VACL, our bandit teacher achieves a higher success rate at test time, since the population size is a discrete variable in nature. Also, in Fig. 4, we observe that the curriculum provided by COST is effective in exploring the task space as agents become increasingly competent.

In the experiments on GRF, we do not include VACL in our baselines in the GRF, since the implementation in the source code of VACL is heavily based on prior knowledge of specific scenarios,

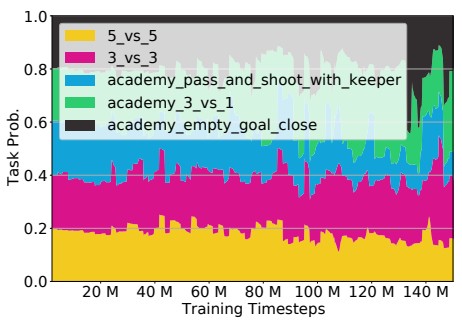
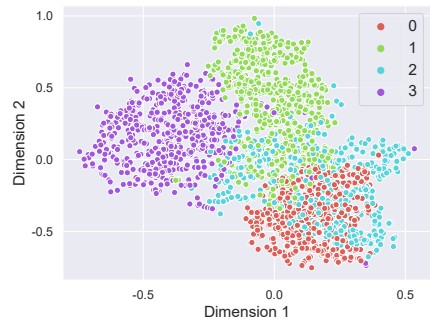

(a) The task distribution of COST during training.

(b) The visualization of contexts

Figure 6: Visualization of Learned Curriculum.

such as the threshold to divide the learning process. We can see that COST has higher win rate and goal difference than IPPO with uniform task sampling in the 5vs5 football competition. The experiments on MPE and GRF show that when the teacher is rewarded by the student's performance, the bandit-based teacher can exploit the student learning stage and give the suitable training tasks to the student.

For ablation study, we replace our contextual multi-armed bandit teacher with uniform task sampling and remove the hierarchical part in the student framework. As shown in Figs. 5a, 5b, we can clearly see that COST can achieve a higher win rate and a greater score difference than COST with uniform and COST w/o. HRL. Also, COST with uniform task sampling outperforms IPPO with uniform task sampling. The difference between these two methods is only the introduction of HRL. It shows the contribution of HRL in the 5vs5 football competition. When removing HRL and contextual multi-armed bandit, the performance degradation w.r.t. COST are similar. It shows that HRL and the contextual multi-armed bandit seem to contribute equally. This can again justify the need for a curriculum learning scheme. However, we can see that COST w. uniform has a larger variance in performance than COST w/o. HRL. It means that uniform sampling might introduce more undesired tasks for student training.

## 5.4 Visualization of Learned Curriculum

We visualize the distribution of task sampling of COST during training based on a selected trial as shown in Fig. 6a. An interesting observation is that the task probability seems nearly uniform. We interpret this into an anti-forgetting mechanism. We can see that at the beginning of training, the task probability seems to be near-uniform, since the teacher should explore the task space and try to keep track of the student's learning status. During training, the probabilities vary over time steps. For example, at about 80-100 million timesteps, we can see a sudden drop in `academy_empty_goal_close` and `academy_3_vs_1_with_keeper`, since the student almost handles the skills learned in such scenarios. However, when training is continued, we can still observe that agents are trained on these tasks more frequently.

We also visualize the distribution of contexts in Fig. 6b using t-SNE [62]. The contexts are collected and stored in a buffer. We divide the contexts into four classes according to the index. We can clearly see a shift in student policy representation from the beginning of training to the end.

## 6 Conclusion

In this paper, to address the scalability and sparse reward issue in the current multi-agent system, we introduce a novel ACL algorithm, Curriculum Oriented Skills and Tactics (COST), to learn complex behaviors from scratch. Specifically, to handle the varying number of agents, we incorporate a population-invariant multi-agent communication framework and exploit a hierarchical scheme for each agent to learn skills to deal with sparse rewards. Moreover, to mitigate the non-stationarity, we model the teacher as a contextual bandit, where the context is represented by the student's policy representation. Empirical results show that our method achieves state-of-the-art performance on several tasks in the multi-particle environment and the challenging 5vs5 competition in GRF.

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

# A Algorithm

---

**Algorithm 2** A contextual bandit algorithm for a small number of contexts

---

**Initialization:** For each context $x$, create an instance $\text{Exp3}_x$ of algorithm Exp3

**for** each round **do**

1. Invoke algorithm $\text{Exp3}_x$ with $x = x_t$

2. Play the action chosen by $\text{Exp3}_x$

3. Return reward $r_t$ to $\text{Exp3}_x$

---

# B Proof of Theorem 4.3

**Theorem 4.3.** *Consider the Lipschitz contextual bandit problem with contexts in $[0, 1]$. The Alg. 1 yields regret $\mathbb{E}[R(T)] = O\left(T^{2/3}(LK \ln T)^{1/3}\right)$.*

*Proof.* Let $S_m$ be the $\epsilon$-uniform mesh on $[0, 1]$, that is, the set of all points in $[0, 1]$ that are integer multiples of $\epsilon$. We take $\epsilon = 1/(d-1)$ where the integer $d$ is the number of points in $S_m$, which will be adjusted later in the analysis.

We apply Alg. 2 to the context space $S_m$. Let $f_{S_m}(x)$ be a mapping from context $x$ to the closest point in $S_m$:

$$f_{S_m}(x) = \min\left(\underset{x' \in S_m}{\operatorname{argmin}} |x - x'|\right)$$

In each round $t$, we replace the context $x_t$ with $f_{S_m}(x_t)$ and call $\text{Exp3}_S$. The regret bound will have two components: the regret bound for $\text{Exp3}_S$ and (a suitable notion of) the discretization error. Formally, let us define the "discretized best response" $\pi^*_{S_m} : \mathcal{X} \to \Phi$: $\pi^*_{S_m}(x) = \pi^*\left(f_{S_m}(x)\right)$ for each context $x \in \mathcal{X}$.

We define the total reward of an algorithm Alg is $\text{Reward}(\text{Alg}) = \sum_{t=1}^{T} r_t$. Then the regret of $\text{Exp3}_S$ and the discretization error are defined as:

$$R_S(T) = \text{Reward}\left(\pi^*_S\right) - \text{Reward}\left(\text{Exp3}_S\right)$$
$$\text{DE}(S) = \text{Reward}\left(\pi^*\right) - \text{Reward}\left(\pi^*_S\right).$$

It follows that regret is the sum $R(T) = R_S(T) + \text{DE}(S)$. We have $\mathbb{E}\left[R_S(T)\right] = \mathcal{O}(\sqrt{TK \log K})$ from Lemma 4.2, so it remains to upper bound the discretization error and adjust the discretization step $\epsilon$.

For each round $t$ and the respective context $x = x_t$, $r\left(\pi^*_S(x) \mid f_S(x)\right) \geq r\left(\pi^*(x) \mid f_S(x)\right) \geq r\left(\pi^*(x) \mid x\right) - \epsilon L$. The first inequality is determined by the optimality of $\pi^*_S$ and the second is determined by Lipschitzness. Summing this up over all rounds $t$, we obtain $\mathbb{E}\left[\text{Reward}\left(\pi^*_S\right)\right] \geq \text{Reward}\left[\pi^*\right] - \epsilon LT$.

Thus, the regret is that

$$\mathbb{E}[R(T)] \leq \epsilon LT + O\left(\sqrt{\frac{1}{\epsilon}TK \log T}\right) = O\left(T^{2/3}(LK \log T)^{1/3}\right) \tag{4}$$

For the last inequality, we choose $\epsilon = \left(\frac{K \log T}{TL^2}\right)^{1/3}$. □

# C 11vs11 Full Game on GRF

We further conduct experiments on the 11vs11 scenario of GRF. As shown in Fig. 7, COST achieves about 50% win rate after training with 200 million timesteps.

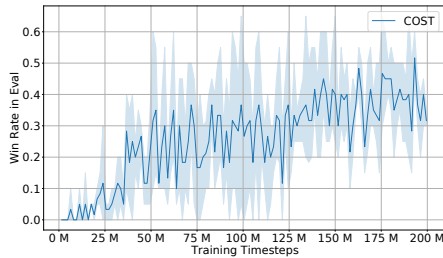

Figure 7: The performance of COST on the 11v11 scenario.

# D   Implementation Details

Here we describe the COST framework. We use the open-sourced Ray RLlib implementation of Proximal Policy Optimization (PPO), which scales out using multiple workers for experience collection. This allows us to use a large amount of rollouts from parallel workers during training to ameliorate high variance and aid exploration. We do multiple rollouts in parallel with distributed workers and use parameter sharing for each agent. The trainer broadcasts new weights to the workers after their synchronous sampling. We now turn our attention to environment-specific settings.

## D.1   Google Research Football

We set five tasks for training the 5vs5 scenario. They are `academy_empty_goal_close`, `academy_pass_and_shoot_with_keeper`, `3_vs_3`, `academy_3_vs_1_with_keeper`, `5_vs_5`. In all scenarios, we do not control our team's goalkeeper.

In the `academy_empty_goal_close`, one agent need to move forward and shoot with an empty goal. In `academy_pass_and_shoot_with_keeper` and `3_vs_3`, two agents are controlled to play against a goalkeeper and 3 players respectively. In `academy_3_vs_1_with_keeper`, three agents are controlled to play against a center-back and a goalkeeper. In `5_vs_5`, 4 agents are controlled to play against 5 players. Without loss of generality, we initialize all player with fixed positions and roles as center midfielders.

We use both MLP and self-attention mechanism for the high-level policy, and use MLP for the low-level policy. For high-level policy, the input is first projected to an embedding using 2 hidden layers with 256 units each and ReLU activation, which is then fed into multi-head self-attention (8 heads, 64 units each). The output is then projected to the actions and values using another fully connected layer with 256 units. For low-level policy, we use MLP with 2 hidden layers with 256 units each, i.e., the default configuration of policy network in RLlib.

Table 2: COST hyper-parameters used in GRF.

| Name | Value |
| --- | --- |
| Discount rate | 0.99 |
| GAE parameter | 1.0 |
| KL coefficient | 0.2 |
| Rollout fragment length | 1000 |
| Training batch size | 100000 |
| SGD minibatch size | 10000 |
| # of SGD iterations | 60 |
| Learning rate | 1e-4 |
| Entropy coefficient | 0.0 |
| Clip parameter | 0.3 |
| Value function clip parameter | 10.0 |

## D.2 MPE

In this environment, agents must cooperate through physical actions to reach a set of landmarks. Agents observe the relative positions of other agents and landmarks, and are collectively rewarded based on the proximity of any agent to each landmark. In other words, the agents have to 'cover' all of the landmarks. Further, the agents occupy significant physical space and are penalized when colliding with each other. The agents need to infer the landmark to cover and move there while avoiding other agents.

The hyper-parameters of COST in MPE are shown in Table 3. In MPE, hyper-parameters such as rollout fragment length, training batch size and SGD minibatch size are adjusted according to horizon of the scenarios so that policy are updated after episodes are done. We use the same network as in GRF, but with 128 units for all MLP hidden layers. Other omitted hyper-parameters follow the default configuration in RLlib PPO implementation.

Table 3: COST hyper-parameters used in MPE.

| Name | Value |
|---|---|
| Discount rate | 0.99 |
| GAE parameter | 1.0 |
| KL coefficient | 0.5 |
| # of SGD iterations | 10 |
| Learning rate | 1e-4 |
| Entropy coefficient | 0.0 |
| Clip parameter | 0.3 |
| Value function clip parameter | 10.0 |

