# OpenReview forum: "Towards Skill and Population Curriculum for MARL"
_NeurIPS.cc/2022/Conference — NeurIPS 2022 Submitted_

### Official Review · Reviewer_1MvZ · 2022-07-09

**Rating:** 4
**Confidence:** 3
**Soundness:** 2 fair
**Presentation:** 3 good
**Contribution:** 1 poor

**Summary:**

The paper proposes a multiagent curriculum method where a "Teacher" proposes tasks to a "Student" to learn to solve. The teacher follows a bandit algorithm to propose tasks taking into account the non-stationarity comes from the student learning algorithm, and the student uses a sophisticated attention-based learning architecture.

**Questions:**

- How is the "distribution of tasks" defined? How is the original (final) task modified to build this distribution?
- How are the policies transferred/adapted across tasks? How similar should the tasks be for the method to work?

**Limitations:**

- The paper has an interesting choice of works added to its references. Arxiv papers [5,6,7] are added to cite the general concepts "reward shaping", "curriculum learning", and "learning from demonstrations", where there are plenty of very comprehensive peer-reviewed surveys in all of those areas. In special, the main surveys of the area of the paper (Curriculum Learning for RL) are not even mentioned (the first is probably the most comprehensive survey while the second is probably the first one to explicitly survey on Curriculum Learning for RL)

Narvekar, Sanmit, et al. "Curriculum Learning for Reinforcement Learning Domains: A Framework and Survey." Journal of Machine Learning Research 21.181 (2020): 1-50.

Da Silva, Felipe Leno, and Anna Helena Reali Costa. "A survey on transfer learning for multiagent reinforcement learning systems." Journal of Artificial Intelligence Research 64 (2019): 645-703.

Perhaps due to an insufficient coverage of the literature, the authors have not added any heuristic-based curriculum learning method in ter experimental evaluation (e.g., one of the many surveyed on those two papers), and also failed to explicitly address many important aspects of a curriculum learning algorithm.

- One of the most important aspects is how the set of subtasks is defined from the final task. If it is assumed that the "distribution of tasks" is manually defined somehow, there is a very big number of heuristic-based curriculum methods that could be applicable for the empirical evaluation and some of them should be included. If this is somehow autonomously defined, there is a smaller group of applicable methods, and the one that comes to my mind at this moment is the Object-oriented curriculum:

Silva, Felipe Leno Da, and Anna Helena Reali Costa. "Object-oriented curriculum generation for reinforcement learning." Proceedings of the 17th International Conference on Autonomous Agents and MultiAgent Systems. 2018.

Another aspect that is not made clear is exactly how to interrupt the training in one task and proceed to the other. This seems to have been defined in a very ad hoc manner for this paper, but the procedure to do this should have been explicitly added in the algorithm.

Moreover, there are some assumptions followed in the work given that this is a multiagent procedure. It is assumed that all agents require constant and perfect communication between agents, which is not true for most practical multi agent scenarios. This should have been made very clear in the paper.

Finally, it is not very clear to me how the knowledge is transferred across the tasks.

- There should be a little more of evidence that the curriculum method is better than not using curriculum. You could add a baseline where exactly the same learning algorithm is used but without switching the task (training directly on the target task).

**Strengths And Weaknesses:**

Albeit the paper explores a very relevant research topic, the experimental evaluation is very limited for a mainly-empirical approach. Moreover, there are a number of aspects and assumptions in the algorithm that are not made clear. Finally, there is a huge group of curriculum for RL papers that are not even mentioned (and hence it's no surprise that those methods are not included in the experimental evaluation). Therefore, I am not entirely convinced the method performs better than other state-of-the-art works (or even classical ones). More details on the reasons in the "Limitations" field.

---

> ### Author Response · Authors · 2022-08-01
> **Responses to your questions and comments**
>
> We thank the reviewer for the valuable feedback to improve our paper. We are confident that the points you make are addressed both in the revisions to the paper made based on your feedback, and by the responses below. We hope this alleviates any concerns you might have, and that you will be prepared to support the paper or further explain what stands between the paper and a supportive assessment on your part.
>
> **Question 1: Distribution of tasks**
>
> We model the teacher with multi-armed bandits, so the distribution of tasks is the probability of different arms. The teacher agent selects from 5 tasks, as shown in lines 598-600 in Appendix D.1. The initial distribution is a uniform distribution, and the final distribution is learned by the algorithm.
>
> **Question 2: How are the policies transferred/adapted across tasks?**
>
> First, we clarify that different tasks in the curriculum are involved with different numbers of agents. For end-to-end training, COST can be trained with varying numbers of agents without stopping and manual resetting. COST just keeps their weights when switching to a new task. The key component to support this feature is a self-attention architecture.
>
> **Question 3: How similar should the tasks be for the method to work?**
>
> These tasks should have the same state/action space for each agent. If some tasks are not helpful to the target task, COST will decrease the probabilities of those tasks.
>
> **Limitation 1: Related work**
>
> In the current version, we mainly cite the auto curriculum learning in MARL. We will include a more comprehensive survey and related work from the heuristic-based curriculum learning.
>
> **Limitation 2: The set of subtasks**
>
> The set of subtasks is set initially. Such subtasks are different in terms of the number of agents, the initial position of agents, and the reward for each agent, which is learned to be set during training automatically. An object-oriented curriculum requires Object-Oriented representation, which needs the similarity between different tasks. This method still needs some human supervision. In COST, the reward of the teacher can direct the exploration in curriculum selection. Moreover, we mainly focus on the auto curriculum learning in MARL. Thus, we can learn how to choose tasks automatically. However, we are glad to discuss the mentioned papers in the rebuttal revision.
>
> **Limitation 3: The training in one task and proceed to the other**
>
> See Question 2. First, we clarify that different tasks in the curriculum are involved with different numbers of agents. For end-to-end training, COST can be trained with varying numbers of agents without stopping and manual resetting. COST just keeps their weights when switching to a new task. The key component to support this feature is a self-attention architecture.
>
> **Limitation 4: Communication**
>
> We mainly focus on the auto curriculum learning in MARL. More practical assumptions about communication are orthogonal to COST.
>
> **Limitation 5: The knowledge is transferred across the tasks**
>
> The knowledge transferred is the basic cooperation via communication and the skill learned by shared low-level policy. We add the experiments about COST without teacher. It achieves ~10% win rate and ~-0.5 goal difference at 100M steps. We will add this ablation in the rebuttal revision.
>
> We hope this response can help you better understand our paper. More discussions on our paper are always welcome!
>
> Best regards,
>
> Paper5108 Authors

---

> > ### Comment · Reviewer_1MvZ · 2022-08-09
> > **Response**
> >
> > Thanks for addressing my comments. I am slightly increasing the evaluation trusting that the authors will perform the changes mentioned in case the paper is accepted.
> >
> > However, the lack of comparison against other state-of-the-art curriculum methods is still a major weakness (2nd only to the unclarity that I hope the authors will fix).
> >
> > Given the number of Curriculum Learning papers mentioned in the two surveys above, the omission of an experimental evaluation against one or two state of the art methods can be read in two ways. 1) The authors didn't know about the other methods and therefore didn't compare against them to make sure the best results so far are achieved; 2) the authors present a very niche and specific setting that is unlikely to be of practical use to other people (and hence it has not been explored despite the relatively big body of works in the area).
> >
> > I do believe it's the former not the latter therefore I recommend to the authors to reevaluate the other curriculum learning methods to see which of them can be implemented in your environment so that the results can be compared (even if some minor adaptations are needed).

---

### Official Review · Reviewer_Vwt6 · 2022-07-09

**Rating:** 4
**Confidence:** 4
**Soundness:** 2 fair
**Presentation:** 3 good
**Contribution:** 2 fair

**Summary:**

This paper proposes a contextual bandit policy based curriculum learning method and applies it to MARL problems. To handle the changing input size induced by the changing numbers of agents, they utilize a self-attention mechanism in the communication channel. To mitigate the non-stationarity, they propose the imitation function to represent the policies and use a cluster algorithm to discretize and reduce the contextal space. The paper is well written and the experiments are introduced clearly.

**Questions:**

1. Section 5.2: it says “The results illustrate that in a simple environment where the student can directly learn to complete the task, there is no need to apply curriculum learning.”, however, the coverage rate in Figure 3 is pretty low (around 0.35 on SimpleSpread, and 0.30 on Push ball). It seems the ACL methods do not help very much to boost the performance, which is also consistent with the generated curriculum in Figure 4 where the curriculum from COST is only around the target task.
2. Figure 6a shows the task distribution generated by COST, which is close-uniform. Why does it produce a very different performance from the uniform curriculum in Figure 5(a)?
3. How much does the HRL policy used in COST help to improve the performance?

**Limitations:**

1. The motivation and advantage of the hierarchical policy and clustering methods are not analyzed clearly. How can they help?
2. The performance on MPE tasks does not show the effectiveness of the proposed method; and the performance on GRF cannot be connected well to the generated task distribution.

**Strengths And Weaknesses:**

Strengths:
1. The researched problem regarding curriculum learning on MARL is interesting and can be beneficial for the community.
2. The proposed using a contextual bandit policy on curriculum learning is promising.
3. The experiments are also well designed.

Weaknesses:
1. The self-attention based population-invariant architecture is similar to the architecture proposed by EPC, which might not be claimed as one of the main contributions.
2. The motivation and benefits of (hierarchical reinforcement learning) HRL in this work are not well explained.
3. The performance of the method seems not significant, especially on MPE tasks. And on both MPE and GRF tasks, the generated task distribution is hard to explain and connected to their corresponding performance.

---

> ### Author Response · Authors · 2022-08-01
> **Responses to your questions and comments**
>
> We thank the reviewer for the very helpful feedback to improve our paper. We are confident that the points you make are addressed both in the revisions to the paper made based on your feedback, and by the responses below. We hope this alleviates any concerns you might have, and that you will be prepared to support the paper or further explain what stands between the paper and a supportive assessment on your part.
>
> **Weakness 1: self-attention in EPC**
>
> Self-attention in EPC is used in the critic to calculate individual Q-value. However, self-attention in COST is used in the actor for agents to communicate. The self-attention in EPC is actually orthogonal to the self-attention in COST.
>
> **Weakness 2: HRL**
>
> The objective of COST is to solve complex cooperation with sparse reward in MARL, which has been stated in the first two sentences of the Abstract and the first paragraph of the Introduction. During the attempt to solve such problems, we find existing promising methods like IPPO, and QMix cannot handle such tasks directly. By introducing curriculum learning, we find IPPO can learn a good policy by jointly training in several simple tasks, which is shown in research question 1 in Experiments and in Fig. 5(a). Although IPPO is a simple but efficient method, the agents trained by IPPO in GRF have similar behaviors. To encourage an agent to learn diverse but coordinated strategies, we introduce a communication and skill framework, as shown in research question 2. We will add this statement in the rebuttal revision.
>
> **Weakness 3: Experiments**
>
> See Question 1 and 2.
>
> **Question 1: boost the performance in MPE**
>
> The objective of COST is to solve complex cooperation with sparse reward in MARL. The MPE is modified with discrete reward, where agents can get a reward only when occupying significant physical space (shown in line 617 in Appendix D.2). That is the reason why the coverage rate in Fig. 3 is low.
>
> In Fig.3, COST performs ~10% better than VACL and IPPO, which is also consistent with the statement in Section 5.2 that there is no need to apply curriculum learning in a simple environment.
>
> **Question 2: Nearly uniform curriculum**
>
> As stated in section 5.4, we interpret the nearly uniform curriculum as an anti-forgetting mechanism. Since COST w. uniform can already achieve ~60% win rate. It shows that some tasks can be inter-independent and parallel (as stated in lines 142-143). Also, we want to point out that the task distribution is compressed by the timestep axes. So, it is visually nearly uniform. We can see ~130M, and the probability of academy_empty_goal_close can be ~40%, which is no more nearly uniform.
>
> **Question 3: HRL contribution**
>
> We already show the contribution of HRL in Fig. 5 as the green line. Also, COST with uniform task sampling outperforms IPPO with uniform task sampling. The difference between these two methods is only the introduction of HRL. These results can be seen in line 324-334.
>
> We hope this response can help you better understand our paper. More discussions on our paper are always welcome!
>
> Best regards,
>
> Paper5108 Authors

---

> > ### Comment · Reviewer_Vwt6 · 2022-08-08
> > **Response to author rebuttal**
> >
> > Dear authors,
> >
> > Thanks for your careful responses which are valuable. However, I still have concerns regarding the weakness above:
> >
> > > Weakness 1: self-attention in EPC
> >
> > >Self-attention in EPC is used in the critic to calculate individual Q-value. However, self-attention in COST is used in the actor for agents to communicate. The self-attention in EPC is actually orthogonal to the self-attention in COST
> >
> > Nonetheless, this contribution is far from being significant in my opinion.
> >
> > > Question 1: boost the performance in MPE
> >
> > In Figures 3a and 4a, the generated numbers of agents by COST is near 16 constantly, however, the IPPO using target 16 agents has much worse performance. Why? And for Figure 3a itself, COST produces a "very good" starting point but cannot significantly improve it anymore. Even though the setting of discrete rewards is hard, it does not mean the task would not need curriculum learning as the coverage rate is still low.
> >
> > > Question 2: Nearly uniform curriculum
> >
> > The explanation of the emergent nearly uniform curriculum and the corresponding better performance than the "true uniform" curriculum is unconvincing, which seems to need more careful comparison and analysis.
> >
> > > Question 3: HRL contribution
> >
> > How different are in the neural networks of IPPO w. and w/o. HRL? Could it be possible that the performance improvement is due to a bigger neural network of HRL policies?
> >
> > In all, I like the idea and the proposed method, which is promising. However, I think the curriculum on the number of agents can significantly affect the dynamics of the environment (also underlying MARL algorithms) in a complicated way and the current experiments and analysis need to be further polished to be a high-quality paper.

---

### Official Review · Reviewer_mgEj · 2022-07-10

**Rating:** 4
**Confidence:** 4
**Soundness:** 3 good
**Presentation:** 3 good
**Contribution:** 2 fair

**Summary:**

This paper proposes a curriculum learning framework for MARL to solve large-scale and more difficult multi-agent tasks. To deal with the problem of varying agent numbers, sparse reward, and the non-stationarity from changing student policies, the authors combine several techniques including the Transformer-based communication structure, hierarchical network design, and contextual bandit for better multi-task learning performance. The proposed method shows good performance on several tasks from MPE and GRF and the visualization results prove that the teacher model can learn a good curriculum for students.

**Questions:**

1. What does the forgetting problem mentioned in Line 40 mean since I thought curriculum learning does not require solving old tasks as continual learning
2.  I didn't quite understand the experimental settings in GRF since only the 5_vs_5 scenario is trained, what is the teacher agent selecting from?
3. Is the modification of baselines and environments like adding the shooting reward necessary? Since the VACL is the only curriculum-based method but it's modified and the original 5_vs_5 scenario can already be solved by some works.




**Limitations:**

Limitations should be discussed more adequately in the paper. Societal impact is likely minimal.

**Strengths And Weaknesses:**

Strengths:
1. This paper attempts to solve the curriculum learning problem in MARL in a multi-task learning way, which is very interesting and intuitive. While most other works only focus on generating the curriculum or learning from a manually selected task sequence, the proposed method applies a teacher-student structure to solve these two problems concurrently.
2. Although there are some previous works utilizing the techniques like the self-attention module or the hierarchical structure of agents in MARL, it is natural to implement them under the curriculum learning setting in this paper and yield good results.
3. The experiment results on selected tasks outperform other baselines on two benchmarks. The ablation study and visualization results are also helpful for understanding the impact of each part.

Weaknesses:
1. The problem of varying numbers of agents in MARL has been widely discussed in MARL and several methods have been proposed and achieved good performance. This paper did not review these works or set them as baselines to compare.
2. The tasks selected in MPE and GRF are mostly 16 agents and 4 agents, which is either not very difficult or really large-scale, making the significance of this work not very good.
3. The baselines didn't include recent works and are clearly not enough for evaluation. The final performance is not impressive and only three scenarios are presented.

---

> ### Author Response · Authors · 2022-08-01
> **Responses to your questions and comments**
>
> We thank the reviewer for very insightful feedback to improve our paper. We are confident that the points you make are addressed both in the revisions to the paper made based on your feedback, and by the responses below. We hope this alleviates the concerns you might have, and that you will be prepared to support the paper or further explain what stands between the paper and a supportive assessment on your part.
>
> **Weaknesses 1: Related work**
>
> We discussed methods with varying numbers of agents in MARL in the second paragraph of the introduction, such as DyMA-CL, EPC, and VACL, and compare with VACL. If we miss other papers, we hope the reviewer can give some references.
>
> **Weaknesses 2: The tasks selected in MPE and GRF**
>
> The MPE is modified with discrete reward, where agents can get a reward only when occupying significant physical space (shown in line 617 in Appendix D.2). In this case, the agents can hardly learn a good policy. The number of 16 agents is much more difficult for algorithms to learn.
>
> In GRF, most related work only study academic scenarios with 2 agents. 5v5 is a much harder scenario. We also show COST’s performance in 11v11 in Fig. 7 in Appendix C.
>
> **Weaknesses 3: The experiments**
>
> We set 3 research questions and choose 2 baselines for each question. The reason why we choose MPE is for a fair comparison with SOTA methods. For GRF, we pick the challenging 5v5 scenario instead of other academic scenarios. Since such academic scenarios, such as academy empty goal close, academy pass and shoot with a keeper, 3 vs 3, academy 3 vs 1 with a keeper, can be easily tackled by IPPO in a few training hours (based on our open-source implementation in the URL). The objective of COST is to solve complex cooperation with sparse reward in MARL, which has been stated in the first two sentences of the Abstract and the first paragraph of the Introduction. So, we focus on the challenging 5v5 scenario.
>
> **Question 1: The forgetting problem**
>
> The forgetting problem means when the agent achieves a high reward in task 1 and then moves to task 2, the agent might not have a similar performance when going back to task 1. We also have some explanations in lines 141-143 and in Section 5.4. Here in our paper, our objective is not to solve the old task, we want to maximize the return on the target tasks as shown in Eq. (1).
>
> **Question 2: only 5v5 scenario**
>
> See Weakness 3. Since such academic scenarios, such as academy empty goal close, academy pass and shoot with akeeper, 3 vs 3, academy 3 vs 1 with a keeper, can be easily tackled by IPPO in a few training hours (based on our open-source implementation in the URL). The objective of COST is to solve complex cooperation with sparse reward in MARL, which has been stated in the first two sentences of the Abstract and the first paragraph of the Introduction. So, we focus on the challenging 5v5 scenario.
>
> The teacher agent selects from 5 tasks, as shown in lines 598-600 in Appendix D.1.
>
> **Question 3: Modification**
>
> For GRF, we only modify the environment by adding the shooting reward for all methods. We tested that it is necessary. Without the shooting reward, all agents can only learn to run with the ball. And 5v5 cannot be solved by STOA methods. See Fig.3 in [1]. We also cite this paper in our main text as [51].
>
> For VACL, we remove its centralized critic, which brings a surprisingly good performance in MPE. Such centralized critic is not the contribution of VACL, so we remove it for a fair comparison.
>
> [1] Containerized Distributed Value-Based Multi-Agent Reinforcement Learning. https://arxiv.org/pdf/2110.08169.pdf
>
> We hope this response can help you better understand our paper. More discussions on our paper are always welcome!
>
> Best regards,
>
> Paper5108 Authors

---

> > ### Comment · Reviewer_mgEj · 2022-08-09
> > **Response to author rebuttal**
> >
> > Thanks for the authors’ valuable response. Some of my concerns are solved and explained but still with some weaknesses:
> > 1. Many recent works have applied the attention mechanism to MARL like REFIL[1] and UPDeT[2] to solve the varying number problem. I think there should be a paragraph reviewing these works in the related works.
> > 2. The baselines are not sufficient, especially in MPE. And the curves of COST in fig.3 left even not show an upward trend and just keep the performance from the beginning of training, while we can see VACL keeps improving and is not converged at the end of training. What if extending the training steps?
> >
> > [1] Iqbal, Shariq, et al. Randomized Entity-wise Factorization for Multi-Agent Reinforcement Learning. ICML, 2021.
> > [2] Hu, Siyi, et al. Updet: Universal multi-agent reinforcement learning via policy decoupling with transformers. ICLR 2021.

---

### Official Review · Reviewer_9xYX · 2022-07-11

**Rating:** 3
**Confidence:** 4
**Soundness:** 2 fair
**Presentation:** 2 fair
**Contribution:** 1 poor

**Summary:**

The authors identify several challenges of cooperative multi-agent reinforcement learning (MARL): sparse rewards, non-stationarity, and the more general requirement compared to existing works, of supporting a varying number of agents or agents with varying existing capabilities in the task space of interest. To tackle these challenges, the authors propose COST, a learning algorithm combining several components: a bandit-based task selection technique, a hierarchical skill-based MARL policy, and a self-attention-based message-passing system.

**Questions:**

- What exactly is the task parameterization that is used for varying the environment in each episode and as input to BIRCH for clustering? This is a core detail that does not seem to be clearly detailed anywhere.
- Similarly, why was VACL chosen as a baseline? Further, why was VACL only used as a baseline for the MPE environment and not Google Research Football?
- Can the authors elaborate on both why BIRCH was chosen for clustering and how this clustering method works? These details would provide missing justification for this essential design choice in the curriculum-learning component of their method.
- Why does the starting number of agents for VACL and COST differ so drastically in Figure 4?
- Are the differences between the learning curves in Figure 5 statistically significant?
- Do the baseline methods also make use of a message-passing self-attention architecture? If not, isn’t this a confounder for the experimental results?

**Limitations:**

COST assumes the ability to use a cheap talk channel both at training and at evaluation, which may often be infeasible in practice. Furthermore, it is not clearly discussed in the paper how scalable this method is to larger numbers of agents or what population size is ideal for applying this method. The authors should clarify these points in a future version of the paper.

**Strengths And Weaknesses:**

Strengths:

- The paper provides a detailed discussion of the challenges of learning complex task spaces in cooperative MARL, and the authors propose several ideas for addressing these issues.
- The self-attention message-passing architecture is an interesting method for supporting an a flexible number of participating agents. This approach, however, is quite similar to that taken by deep coordination graphs [1], and so a discussion and comparison with respect to this line of work seems necessary.

Weaknesses:

- While there are good ideas in this paper, the current presentation of the method and results is disorganized and unfocused. The result is that while the authors identify several important challenges for improving cooperative MARL, they fail to provide much insight into whether any of their proposed solutions are at all effective. This paper is trying to do too much at once, and as a result, focuses on too many problems simultaneously with independent methods for each, while not clearly demonstrating the benefits of their method for solving any one of them.
- The experiments show ablations only on the Google Research Football environment, and only for the curriculum and hierarchical skill-based components of the method. Meanwhile, the results compare to baselines that seem to use a completely different policy architecture, as the baselines do not seem to use the self-attention message-passing components. As a result, it seems that the self-attention architecture acts as a confounder for the results, making it hard to draw any conclusions from the presented results.
- In procedurally-generated environments, it is important to test for generalization to held-out test configurations of the environment. Some methods that use a return-maximizing curriculum, as performed by COST's multi-arm bandit, may result in overfitting to the training tasks. This aspect of their problem setting is completely ignored, and fully missing from the paper.
- Some design decisions seem entirely arbitrary without clear motivation, for example, the choice of BIRCH for task clustering and the decision to train the higher-level skill policies independently, while the low-level policies with parameter sharing across all agents.
- The curriculum visualization results seem to suggest no real curriculum is being learned. This also seems supported by the large overlapping error bars between COST and the COST without curriculum ablation in the Google Research Football learning curves.
- The authors miss crucial related work from the unsupervised environment design subfield of reinforcement learning [1,2,3,4] as wells prior work on Teacher-Student Curriculum Learning [5].
- The authors should consider moving the multi-armed bandit section in Section 2, their related works section to Section 3, which serves as their background section.
- Overall, the paper presents a large number of ultimately disparate ideas for improving cooperative MARL, without providing sufficient experiments teasing apart the effect of each newly proposed component. Each of the proposed solution components also already exists in a similar form in prior works (see cited references below), while, judging from the experimental results, no clear emergent benefits seem to result from their combination. This suggests that the novelty of this paper is limited.

References:

[1] Böhmer et al, 2020. Deep Coordination Graphs.

[2] Dennis et al, 2020. Emergent Complexity and Zero-shot Transfer via Unsupervised Environment Design.

[3] Jiang et al, 2021. Prioritized Level Replay.

[4] Jiang et al, 2021. Replay-Guided Adversarial Environment Design.

[5] Parker-Holder et al, 2022. Evolving Curricula with Regret-Based Environment Design.

[6] Matiisen et al, 2019. Teacher-Student Curriculum Learning

---

> ### Author Response · Authors · 2022-08-01
> **Responses to your questions and comments (Part 1)**
>
> We thank the reviewer for the very helpful feedback to improve our paper. We are confident that the points you make are addressed both in the revisions to the paper made based on your feedback, and by the responses below. We hope this alleviates the concerns you might have, and that you will be prepared to support the paper or further explain what stands between the paper and a supportive assessment on your part.
>
> **Strengths: Deep coordination graphs**
>
> From the perspective of dealing with huge numbers of agents, DCG employs (1) parameter sharing among agents, and (2) value factorization/decomposition with pairwise payoffs and individual utilities. Whilst, COST employs (1) parameter sharing among agents, and (2) self-attention message-passing. DCG focuses on the critic while COST focuses on the actor. So COST is orthogonal to DCG.
>
> Also, the objectives of DCG and COST are different. DCG aims to solve relative overgeneralization, while COST aims to solve complex cooperation with sparse reward. Notice that value decomposition methods cannot be directly applied in sparse reward scenarios, as shown in Fig.5 (a). QMix hardly wins the opponents.
>
> **Weakness 1: Disorganized and unfocused**
>
> The objective of COST is to solve complex cooperation with sparse reward in MARL, which has been stated in the first two sentence of Abstract and first paragraph of Introduction. During the attempt to solving such problems, we find existing promising methods like IPPO, QMix cannot handle such tasks directly. By introducing curriculum learning, we find IPPO can learn a good policy by jointly training in several simple tasks, which is shown in research question 1 in Experiments and in Fig. 5(a). Although IPPO is a simple but efficient method, the agents trained by IPPO in GRF have similar behaviors. To encourage an agent to learn diverse but coordinated strategies, we introduce communication and skill framework, as shown in research question 2.
>
> **Weakness 2: Experiments**
>
> - **Ablations only on the Google Research Football environment.**
>
> 5v5 scenario is much more complex than MPE, where each experiment requires a node with 128-core CPU and 4 A100 GPUs to run for 1-2 days (as shown in line 281-282) (based on Industry-Grade open-source library Ray RLlib). The ablation on GRF could show how each component in COST contributes in such complex environments.
>
> - **Missing ablation on self-attention architecture.**
>
> First, we clarify that different tasks in the curriculum are involved with different numbers of agents. For end-to-end training, COST is trained with varying numbers of agents without stopping and manual resetting. The key component to support this feature is a self-attention architecture. When removing the self-attention architecture, COST degrades to independent hierarchical PPO with curriculum learning.
>
> Then, we add the ablation on self-attention architecture. By removing the self-attention architecture, it achieves a win-rate of 62%±7% and 1.8±0.5 goal difference. The performance is slightly lower than COST w/o HRL. It shows that self-attention contributes more than HRL since agents can exchange information for better cooperation. We will add this ablation in the rebuttal revision.
>
> **Weakness 3: Held-out test configurations**
>
> Generalization is the purpose of many methods in procedurally generated environments. However, the purpose of COST is to overfit the test environment with much more difficulty. As shown in Eq. (1), we want to maximize the return on the target tasks instead of the return on some distribution of various tasks.
>
>
> **Weakness 4: Arbitrary design decisions**
>
> - **the choice of BIRCH**
>
> Since the new contexts are added to the buffer continuously, we need an *online* cluster algorithms. BIRCH is one of the most popular online cluster algorithms, which has a good implementation by scikit-learn [1]. We use this implementation directly for reproducibility.
>
> - **training scheme of HRL**
>
> By partially sharing network structure, agents can learn diverse but coordinated strategies [2]. To encourage an agent to learn diverse but coordinated strategies, we introduce communication and skill framework, as shown in research question 2.

---

> > ### Author Response · Authors · 2022-08-02
> > **Responses to your questions and comments (Part 2)**
> >
> > **Weakness 5: The curriculum visualization**
> >
> > As stated in section 5.4, we interpret the nearly uniform curriculum to an anti-forgetting mechanism. Since COST w. uniform can already achieve ~60% win rate. It shows that some tasks can be inter-independent and parallel (as stated in lines 142-143). Also, we want to point out that the task distribution is compressed by the timestep axes. So, it is visually nearly uniform. We can see ~130M, and the probability of academy_empty_goal_close can be ~40%, which is no more nearly uniform.
> >
> > The overlap is on the (-std) part of COST and the (+std) part of COST w. uniform. The mean win rate and goal difference of COST is higher than the mean+std win rate and goal difference of COST with uniform. As stated in section 5.4, we interpret the nearly uniform curriculum to an anti-forgetting mechanism. Since COST w. uniform can already achieve ~60% win rate. It shows that some tasks can be inter-independent and parallel (as stated in lines 142-143).
> >
> > **Weakness 6: Crucial related work**
> >
> > In the current version, we mainly cite the auto curriculum learning in MARL. We will include the related work from the unsupervised environment design for RL.
> >
> > **Weakness 7: The multi-armed bandit section**
> >
> > Thanks for the suggestion.
> >
> > **Question 1: Task parameterization and input of BIRCH**
> >
> > For GRF, as stated in Appendix D.1, we set five training tasks. The task parameterization is the number of agents, the initial position of agents, and the reward for each agent. For MPE, the task parameterization is the number of agents and the initial position of agents.
> > The input of BIRCH is the context, which is the student’s policy representation. The details are in Section 4.2.1. We also visualize the context in Fig. 6(b) and in Section 5.4
> >
> > **Question 2: VACL chosen as a baseline**
> >
> > We choose VACL since it is the SOTA method by now in the curriculum learning in MARL. And the reason why only used for MPE is that VACL is heavily designed for MPE in their open-sourced code, such as the threshold in their value quantization. For a fair comparison, we only compare it in MPE.
> >
> > **Question 3: BIRCH**
> >
> > See weakness 4. Generally, BIRCH maintains a height-balanced tree structure. BIRCH first scans the whole data and fits them into the clustering feature (CF) trees. There are mainly four phases that are followed by the algorithm of BIRCH: (1) scanning data into memory, (2) condensing data (resize data), (3) global clustering, and (4) refining clusters. In condensing, it resets and resizes the data for better fitting into the CF tree. In global clustering, it sends CF trees for clustering using existing clustering algorithms. Finally, refining fixes the problem of CF trees where the same valued points are assigned to different leaf nodes. We will include the description of BIRCH in the Appendix.
> >
> > **Question 4: The starting number of agents**
> >
> > It is because we plot the mean number of agents during training over different seeds. COST starts from a uniform distribution over the numbers of agents: {2, 4, 8, 16}, as shown in line 287. According to the source code of VACL, VACL starts from 2 agents.
> >
> > **Question 5: Statistically significance**
> >
> > The overlap is on the (-std) part of COST and the (+std) part of COST w. uniform. The mean win rate and goal difference of COST is higher than the mean+std win rate and goal difference of COST with uniform. Based on this, we can calculate \sigma = |mean_1 – mean_2|/\sqrt(std^2/5+std^2/5) = 1.7, and hence P(z>σ) =1−pnorm(1.7)= 0.04457<0.05, which is statistically significant.
> >
> > **Question 6: Self-attention structure**
> >
> > See Weakness 2. We add the ablation on self-attention architecture. By removing self-attention architecture, it achieves a win-rate of 62%±7% and 1.8±0.5 goal difference. The performance is slightly lower than COST w/o HRL. It shows that self-attention contributes more than HRL since agents can exchange information for better cooperation. We will add this ablation in the rebuttal revision.
> >
> > [1] https://scikit-learn.org/stable/modules/generated/sklearn.cluster.Birch.html
> >
> > [2] Celebrating Diversity in Shared Multi-Agent Reinforcement Learning.
> >
> > We hope this response can help you better understand our paper. More discussions on our paper are always welcome!
> >
> > Best regards,
> >
> > Paper5108 Authors

---

> > > ### Comment · Reviewer_9xYX · 2022-08-06
> > > **Response to author rebuttal**
> > >
> > > Many thanks to the authors for their response.
> > >
> > > Regarding the comparison to DCG, you mention that COST focuses on the actor, while DCG focuses on the critic. The advantage of DCG is then that it confroms to the paradigm of centralized training and decentralized execution (CTDE), whereby the agents are trained with information sharing at train time and are executed independently at test time. COST agents are unable to execute independently at test-time due to the shared message passing architecture. This not only makes the method less practical than fully CTDE methods, but it also makes the comparison to fully CTDE methods like QMIX and IPPO unfair.
> > >
> > > Regarding my critique of the paper being unfocused, you mention that
> > >
> > > >To encourage an agent to learn diverse but coordinated strategies, we introduce communication and skill framework, as shown in research question 2
> > >
> > > It is not clear to me from the paper and this subsequent discussion why we should expect communication and HRL to result in more diverse policies. This claim is unsubstantiated by your work, hence my critique of the method having too many unjustified components.
> > >
> > > Further, there still is no clear reasoning provided for the specific design choices around the HRL training setup.
> > >
> > > Given these outlying issues, I choose to keep my current rating of this work.

---

> > > > ### Author Response · Authors · 2022-08-07
> > > > **Re: Response to author rebuttal**
> > > >
> > > > We thank you again for your valuable time and comments on our rebuttal.
> > > >
> > > > **Q1: DCG**
> > > >
> > > > CTDE and learning communication are both common practices in MARL. These two are orthogonal to each other, as shown in [1,2,3] where agents communicate without CTDE and [4,5,6] with CTDE. It is a fact that agents are unable to execute independently at test-time due to the shared message passing architecture, however, it is a common setting in MARL communication methods, which is unfair to be considered as an unpractical point in COST. The objective of COST is to solve complex cooperation with sparse reward in MARL, instead of independent execution at test time.
> > > >
> > > > Besides, IPPO is not a CTDE method, but instead a fully independent method. If removing the curriculum learning part from COST, the differences between COST and IPPO are communication and HRL. We showed in research question 2 about these two parts. We add an ablation study about removing the curriculum learning part from COST (i.e. IPPO with communication and HRL), it achieves a win rate of 10%±2% and 0.5±0.3 goal difference.
> > > >
> > > > **Q2: Communication and HRL & diverse policies & specific design in HRL**
> > > >
> > > > The communication and HRL in MARL are studied a few decades ago [7] and studied separately recently in the ego of deep RL [8,9]. The reason why we introduce communication and HRL is that in IPPO all agents are sharing parameters, which leads to similar behaviors among them. Learning more diverse policies is a well-known challenge in MARL. As shown in [10], “when parameters are shared, agents tend to acquire homogeneous behaviors”, and “for example, the unsatisfactory performance of state-of-the-art MARL algorithms on Google Research Football highlights an urgent demand for diverse behaviors.” In their work, they pointed out that “the shared local Q-function does not have enough capacity to present different policies for each agent.” They “additionally equip each agent i with an individual local Q-function”. In [11], the authors proposed a diversity-driven extensible HRL for discovering inter-task transferable skills. In [12], “agents learn useful and distinct skills at the low level via independent Q-learning, while they learn to select complementary latent skill variables at the high level via centralized multi-agent training”.
> > > >
> > > > Motivated by [10, 13,14], the parameters should be partially shared between agents. In COST, as shown in lines 171-182, the high-level policies of all agents don’t share parameters, while the low-level policies share parameters. By using HRL, the agents learn how to perform each individual subtask, and by communication, they learn how to coordinate with other agents. In Fig. 6, we showed that the contexts, i.e., the student’s policy representation are scattered/diverse and have a shift from the beginning of training to the end.
> > > >
> > > > Best regards,
> > > >
> > > > Paper5108 Authors
> > > >
> > > >
> > > > [1] Learning to communicate with deep multi-agent reinforcement learning. Foerster J, Assael I A, De Freitas N, et al. NeurIPS 2016.
> > > >
> > > > [2] Learning multiagent communication with backpropagation. Sukhbaatar S, Fergus R. NeurIPS 2016.
> > > >
> > > > [3] Learning attentional communication for multi-agent cooperation. Jiang J, Lu Z. NeurIPS 2018.
> > > >
> > > > [4] Learning multi-agent communication through structured attentive reasoning. Rangwala M, Williams R. NeurIPS 2020.
> > > >
> > > > [5] Learning when to communicate at scale in multiagent cooperative and competitive tasks. Singh A, Jain T, Sukhbaatar S. ICLR 2018
> > > >
> > > > [6] Tarmac: Targeted multi-agent communication. Das A, Gervet T, Romoff J, et al. ICML 2019.
> > > >
> > > > [7] Hierarchical Multi-Agent Reinforcement Learning. Makar R, Mahadevan S, Ghavamzadeh M. JAAMAS 2001.
> > > >
> > > > [8] Hierarchical reinforcement learning: A comprehensive survey. Pateria S, Subagdja B, Tan A, et al.  ACM Computing Surveys 2021.
> > > >
> > > > [9] Emergent multi-agent communication in the deep learning era. Lazaridou A, Baroni M. arXiv:2006.02419.
> > > >
> > > > [10] Celebrating Diversity in Shared Multi-Agent Reinforcement Learning. Chenghao L, Wang T, Wu C, et al. NeurIPS 2021.
> > > >
> > > > [11] Diversity-Driven Extensible Hierarchical Reinforcement Learning. Song Y, Wang J, Lukasiewicz T, et al. AAAI 2019.
> > > >
> > > > [12] Hierarchical Cooperative Multi-Agent Reinforcement Learning with Skill Discovery. Yang J, Borovikov I, Zha H. AAMAS 2020.
> > > >
> > > > [13] Diversity is all you need: Learning skills without a reward function. Eysenbach B, Gupta A, Ibarz J, et al. ICLR 2018.
> > > >
> > > > [14] The option-critic architecture. Bacon P L, Harb J, Precup D. AAAI 2017.

---

> > > > > ### Comment · Reviewer_9xYX · 2022-08-08
> > > > > **Response to authors**
> > > > >
> > > > > Of course I agree communication and HRL are important areas of research. That is not the basis of my critique.
> > > > >
> > > > > To reiterate my criticism regarding the centralized communication channel: Your experiments compare a method that uses centralized message-passing at each time step at **test time** and compare to baselines (IPPO and QMIX) that are **fully decentralized** at test time. In this light, it is not surprising that sharing information in this way at test time confers performance gains. Moreover, you do not seem to provide an ablation of your method to assess the individual contribution of this centralized communication channel.
> > > > >
> > > > > My criticism re the HRL component centers on your claim that it should induce more diverse policies. This does not seem well-supported by the facts and arguments presented in the paper.

---

### Author Response · Authors · 2022-08-01
**Joint response to reviewers**

We thank the reviewers for their valueble feedback. We are confident that in responding to individual questions and comments, we can address your main concerns. Below we summarize the main improvements and additional results that we have added to our revision. In addition, we would like to address misunderstandings that concerned multiple reviewers, which we expand upon in a more tailored form in individual responses.

**Motivation**

The objective of COST is to solve complex cooperation with sparse reward in MARL, which has been stated in the first two sentence of Abstract and first paragraph of Introduction. During the attempt to solving such problems, we find existing promising methods like IPPO, QMix cannot handle such tasks directly. By introducing curriculum learning, we find IPPO can learn a good policy by jointly training with several simple tasks, which is shown in research question 1 in Experiments and in Fig. 5(a). Although IPPO is a simple but efficient method, the agents trained by IPPO in GRF have similar behaviors. To encourage an agent to learn diverse but coordinated strategies, we introduce communication and skill framework, as shown in research question 2.

**Why only 5v5 on GRF?**

For GRF, as shown in “Multi-Agent Experiments” reported by the original GRF paper [1], in the 3 vs 1 with keeper scenario, 3 agented trained by 50M steps of training can achieve 0.86 ± 0.08 scores (not the goal difference). Based on our open-sourced implementation in the URL, such academic scenarios, such as academy empty goal close, academy pass and shoot with keeper, 3 vs 3, academy 3 vs 1 with keeper, can be easily tackled by IPPO in a few training hours. However, when we train on 5v5 scenario, we find existing promising methods like IPPO, QMix cannot handle such tasks directly. This result can also be cross-validated in Fig.3 in [2].
The objective of COST is to solve complex cooperation with sparse reward in MARL. So, we focus on the challenging 5v5 scenario.

**Ablation study**

By answering the research question 2 in Section 5.3, we already show the contribution of HRL in Fig. 5 as the green line. Also COST with uniform task sampling outperforms IPPO with uniform task sampling. The difference between these two methods is only the introduction of HRL. These results can be seen in line 324-334. As for the contribution of self-attention structure, we add the ablation on self-attention architecture. By removing self-attention architecture, it achieves a win-rate of 62%±7% and 1.8±0.5 goal difference. The performance is slightly lower than COST w/o HRL. It shows that self-attention contributes more than HRL since agents can exchange information for better cooperation. We will add this ablation in the rebuttal revision.

Overall, we restate our contributions here: (1) a novel auto curriculum learning method for cooperative MARL with sparse reward, (2) a theoretically sound multi-armed bandit method for choosing different tasks, (3) a population-invariant policy structure for varying numbers of training agents, (4) open source the reproducible code for the community (the code button in the URL).

[1] Google Research Football: A Novel Reinforcement Learning Environment. https://arxiv.org/pdf/1907.11180.pdf

[2] Containerized Distributed Value-Based Multi-Agent Reinforcement Learning. https://arxiv.org/pdf/2110.08169.pdf

---

### Meta-Review · Area_Chair_zccd · 2022-08-21

**Recommendation:** Reject
**Confidence:** Certain

**Metareview:**

This paper addresses some important problems in curriculum learning. Reviewers were generally happy with the approach. The main criticisms are around the evaluation. There is a lot of prior and similar work in the the literature, and reviewers felt that the paper lacked strong baselines against state of the art curriculum learning methods (reviewer 1MvZ and mgEj). Reviewer Vwt6 also pointed out issues with evaluation. For example, it was unclear if difference in reported performance was due to differences in algorithm, or in neural net sizes. Including stronger evaluation, and more references to related methods from the literature, would make the paper much stronger. Our conclusion is that the paper is not yet ready for publication.

**Award:**

No

---

### Decision · Program_Chairs · 2022-09-14

Reject